# Reflective Multi-Agent Collaboration based on Large Language Models

**Xiaohe Bo**[1], **Zeyu Zhang**[1], **Quanyu Dai**[2], **Xueyang Feng**[1],
**Lei Wang**[1], **Rui Li**[1], **Xu Chen**[1,*] **Ji-Rong Wen**[1]

[1] Gaoling School of Artificial Intelligence, Renmin University of China
[2] Huawei Noah's Ark Lab

{xiaohe,zeyuzhang,xueyangfeng,wanglei154,lirui121200,xu.chen,jrwen}@ruc.edu.cn,
daiquanyu@huawei.com

## Abstract

Benefiting from the powerful language expression and planning capabilities of Large Language Models (LLMs), LLM-based autonomous agents have achieved promising performance in various downstream tasks. Recently, based on the development of single-agent systems, researchers propose to construct LLM-based multi-agent systems to tackle more complicated tasks. In this paper, we propose a novel framework, named **COPPER**, to enhance the collaborative capabilities of LLM-based agents with the self-reflection mechanism. To improve the quality of reflections, we propose to fine-tune a shared reflector, which automatically tunes the prompts of actor models using our counterfactual PPO mechanism. On the one hand, we propose counterfactual rewards to assess the contribution of a single agent's reflection within the system, alleviating the credit assignment problem. On the other hand, we propose to train a shared reflector, which enables the reflector to generate personalized reflections according to agent roles, while reducing the computational resource requirements and improving training stability. We conduct experiments on three datasets to evaluate the performance of our model in multi-hop question answering, mathematics, and chess scenarios. Experimental results show that COPPER possesses stronger reflection capabilities and exhibits excellent generalization performance across different actor models.

## 1 Introduction

With the emergence of Large Language Models, LLM-based autonomous agents are becoming a research hotspot in the field of artificial intelligence. Leveraging the impressive planning and reasoning ability of LLMs, these agents can understand and generate human-like instructions, engage in sophisticated interactions, and make decisions in a wide range of contexts, leading to remarkable success in various downstream tasks [26, 24, 25, 5, 22]. Recently, based on the development of single-agent systems, researchers propose to construct multi-agent systems in response to the growing task complexity. Prior works [31, 12, 8] suggest that multiple agents can help improve factuality and reasoning, encourage divergent thinking, and effectively facilitate task completion.

To improve the collaborative performance of multi-agent systems, various cooperation frameworks [31, 4, 23, 39] have been developed, which generally encode intricately crafted agent profiles and cooperation mechanisms into prompts. However, hindered by the contextual understanding ability of LLMs, such frameworks fall short of fully exploiting the collaborative capacities of agents. To tackle this challenge, one natural idea is to gather extensive collaborative data for agents' fine-tuning. Yet this strategy risks diminishing the model's general abilities [35], contradicting the aspiration to

---

*Corresponding Author.

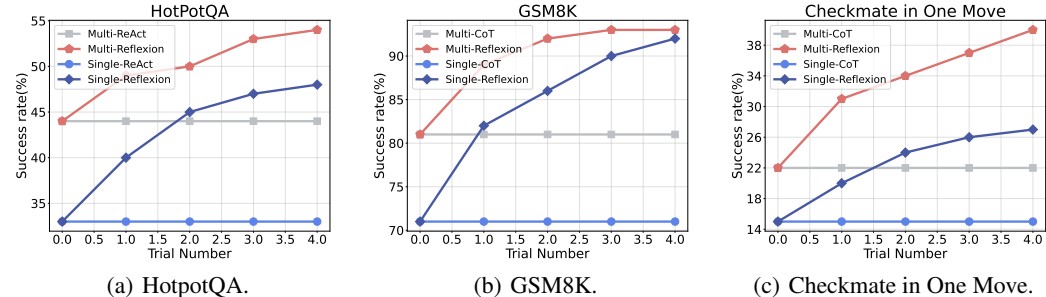

|     |     |     |
| --- | --- | --- |
| (a) HotpotQA. | (b) GSM8K. | (c) Checkmate in One Move. |

Figure 1: Performance comparison between single-agent and multi-agent systems on three datasets. We use GPT-3.5 as the base model and prompts are shown in the Appendix H.

attain artificial general intelligence (AGI). Considering that, in this paper, we propose to optimize the collaboration process through the self-reflection mechanism [27], with which binary or scalar rewards from environments can be converted into verbal reflections, providing additional context to improve task performance. To accommodate the characteristics of multi-agent systems, we additionally incorporate agent profiles in reflector prompts for agent-specific reflections and adopt a fully observable setting to facilitate agents' error detection.

Although the self-reflection mechanism enables iterative refinement, deriving useful reflections from a pre-trained, frozen LLM is challenging. In dealing with that, Retroformer [38] suggests approximating reflection rewards with the difference between two consecutive episode returns and training a plug-in reflector through policy optimization. However, extending the method to multi-agent systems is not straightforward. On the one hand, the episode difference score from the environment can only capture the overall contribution from all agents' reflections, while the credit to each agent's reflection is unknown and nontrivial to obtain. Employing the overall score directly and uniformly for reflector training of multiple different agents could lead to lazy reflectors, and thus the credit assignment problem becomes critical. On the other hand, to achieve personalized reflections of intelligent agents, the number of reflectors to be fine-tuned will expand proportionally with the number of agents in the system, posing challenges for practical applications in real-world scenarios.

To address the challenges above, we propose a reflective framework: **CO**unterfactual **PP**O **E**nhanced Shared **R**eflector for LLM-based Multi-Agent Collaboration, named **COPPER**. For the first challenge, we propose counterfactual rewards as supervision signals for individual agent reflections. Specifically, we first integrate the reflections of all agents into the corresponding actor model prompts and utilize the episode return difference score as the *overall reward*. Then we sequentially marginalize out the reflection of one agent, repeat the interaction process, and attain a new task score. The resulting episode return difference is referred to as *marginal reward*. The *counterfactual reward* is then calculated by subtracting the marginal reward from the overall reward to reflect the contribution of the removed reflection. For the second challenge, considering the homogeneity between different reflectors, which means their action space (reflection) is consistent and the optimization objectives are aligned (to assist in solving the overall task), we propose to train a shared reflector for agents in the collaboration system. With carefully designed prompts, the shared reflector can grasp the role information of each agent, simultaneously reducing the demand for computational resources and enriching the training data pool for stable training. The counterfactual reflection data from all agents are then collected and utilized to train a shared reflector through proximal policy optimization (PPO).

Our contributions can be summarized as follows:

• We propose a novel reflection framework, named COPPER, to improve the multi-agent collaboration. We incorporate agent profiles into reflector prompts for agent-specific reflections and adopt a fully observable setting to assist in error detection.

• We propose to train a shared reflector using our counterfactual PPO mechanism. To alleviate the credit assignment problem in multi-agent systems, we design counterfactual rewards to rate each agent's reflection. Besides, we propose to train a shared reflector, which could generate personalized reflections while reducing computational resource demands and improving training stability.

• Experimental results on three open-source datasets demonstrate that COPPER possesses stronger reflection capabilities against baselines. More concretely, compared to the initial success rate,

COPPER brought improvements of 31.8%, 18.5%, and 86.4% on the HotPotQA, GSM8K, and Checkmate in One Move datasets, respectively.

## 2  Related Work

### 2.1  LLM-based Multi-Agent Systems

Based on the development of single-agent systems, LLM-based multi-agent systems have been rapidly studied and achieved significant progress in complex task resolution and world simulation. Within the task resolution domain, various agents, each with specialized expertise are developed to collaborate on complex problems. For instance, [9, 32, 3, 29, 17] suggest improving the accuracy of scientific question-answering tasks through multi-agent debates. [23, 13] suggest constructing multi-agent systems for software development following the waterfall or Standardized Operating Procedures (SOPs) workflow. Another mainstream application scenario of LLM-based multi-agent systems is the world simulation, which mainly leverages the role-playing abilities of agents to represent different roles and perspectives within a simulated environment. Research in this area is advancing quickly and encompasses a wide variety of fields, including social sciences [42, 11], gaming [33, 34, 18, 20, 1], psychology [2], economics [16, 41], policy making [15], etc. In this paper, we focus on improving the complex problem-solving abilities of multi-agent systems.

### 2.2  Self-Reflection of Large Language Models

Various studies on reflection mechanisms have been proposed, which play a crucial role in enabling LLM-based agents to learn from the environment and improve themselves autonomously. Early works primarily focus on refining responses based on a single feedback [19, 6] or contrast between multiple models [10, 40] and fail to form a comprehensive understanding of the task based on past experiences. Recently, Reflexion [27] involves prior trajectories and environmental rewards to generate reflections, which are further incorporated into the context of subsequent episodes. Although it enables iterative enhancements, the effectiveness of reflections heavily relies on the model's inherent reflective capabilities. In light of that, Retroformer [38] proposes to fine-tune the reflector using environmental rewards with a standard RLHF [21] process. However, optimizing reflection in multi-agent systems remains challenging, which is crucial for improving agents' cooperation capacity and task performance.

## 3  Preliminary

In this paper, we use a tuple $(N, \mathcal{S}, \mathcal{A}, \mathcal{P}_{\xi_o}, \mathcal{R})$ to denote the LLM-based multi-agent cooperation system, where $N$ stands for the number of agents, $\mathcal{S} = S_1 \times S_2 \times \cdots \times S_N$ is the joint space of environment states, $\mathcal{A} = A_1 \times A_2 \times \cdots \times A_N$ is the joint action space and $\mathcal{P}_{\xi_o} : \mathcal{S} \times \mathcal{A} \to \mathcal{S}$ is the state transition function. Here, we denote the randomness associated with the state transition using $\xi_o$ according to [38]. In cooperative settings, all agents share an aligning goal, and the reward function $\mathcal{R} : (\mathcal{S}, \mathcal{A}) \to \mathbb{R}$ is typically designed to promote collaboration. One major challenge in cooperation settings is credit assignment, which means we need to decompose $\mathcal{R}$ into $R_1 \times R_2 \times ... \times R_N$ and evaluate the agent contribution with respect to their objective in the form of a scalar value. The multi-agent systems complete the target task through interactions with the environment. Here we use trajectory $\tau = \{s_0, a_0, s_1, a_1, \cdots, s_T, a_T\}$ to denote the process and describe the accumulative reward using $R(\tau)$, where $R(\tau) = \sum_{t=0}^{T} \mathcal{R}(s_t, a_t)$ and $T$ is the length of the trajectory. In most of the situations, rewards from the environment are sparse, which means $\mathcal{R}(s_t, a_t)$ are mostly zero except very few states, such as the terminal state for indicating task success or failure.

Specifically, for each agent $i$, we consider its actor model as a function $\mathcal{M}^i_{\xi_l} : \mathcal{X}_i \to A_i$, where $\mathcal{X}_i$ is the space of the prompts and $\xi_l$ represents the random variables involved in the sampling process. To maintain the general abilities of agents, in this paper, we select LLMs with frozen parameters such as ChatGPT and GPT-4 as actor models. Current environment states are incorporated into prompts in the form of natural language and actions are selected based on the contextual learning ability of LLMs. Meanwhile, we propose to improve collaboration through self-reflection and introduce a reflector model $\mathcal{M}^i_{\xi_r, \theta} : (\mathcal{T}, \mathbb{R}) \to \mathcal{X}_i$ for each agent, where $\mathcal{T}$ denotes the space of trajectories, $\xi_r$ represents the randomness in the reflector model and $\theta$ denotes the learnable parameters. The reflector model takes the prior trajectory and the reward signal from the environment as input, outputting reflections

to refine the prompt of the corresponding actor model. We fine-tune reflector models through policy optimization for better task performance in specific environments.

## 4 Method

### 4.1 Multi-Agent Collaboration

LLM-based multi-agent systems aim to deliver advanced capabilities by leveraging collective intelligence and specializing LLMs into agents with distinct capabilities. In this paper, each agent in the system operates in a predetermined sequence, taking turns to produce responses, while a shared message pool is maintained to facilitate efficient communication. We illustrate the details of the collaboration settings in Appendix A.

Specifically, in dealing with problem $k$ at time $t$, agent $i$ ($i = t \mod N$) first subscribes the preceding interaction records $[s_{k,i}, a_{k,i}]_{i=0}^{t-1}$ from the message pool, and then acquires the current environment state $s_{k,t}$. The decision-making process can be expressed as:

$$a_{k,t} = \text{Actor}^i(p^i, [s_{k,i}, a_{k,i}]_{i=0}^{t-1}, s_{k,t}), \tag{1}$$

where $p^i$ is the profile of the current agent, which encompasses its role, action space, and additional constraints in the form of natural language. Once reaching a decision, the agent publishes a new message $\{s_{k,t}, a_{k,t}\}$ to the message pool.

However, during implementation, the interaction history could potentially exceed the token limit of LLMs, given the large number of agents and decision steps. To address this challenge, we introduce a context model to recursively update the interaction history from each agent's perspective, serving as its short-term memory. New messages since the agent's last action and the profile will be integrated to form a new short-term memory based on the previous one. The process can be written as:

$$sm_{k,t}^i = \text{Context}^i(p^i, sm_{k,t-1}^i, \{s_i, a_i\}_{i=\max(0,t-N+1)}^t), \tag{2}$$

where $sm_{k,t}^i$ represent the short-term memory of agent $i$ when solving problem $k$ at time $t$.

We then replace the interaction history with the agent's short-term memory. Therefore, the decision-making process can be further rewritten as:

$$a_{k,t} = \text{Actor}^i(p^i, sm_{k,t}^i, s_{k,t}). \tag{3}$$

### 4.2 Multi-Agent Reflection Framework

To bolster the collaborative performance of multi-agent systems in specific scenarios, while preserving the general capabilities of agents, we introduce a self-reflection mechanism to multi-agent systems, of which the details are shown on the left part of Figure 2. Using environmental rewards as guidance, the generated reflections could act as semantic gradient signals by providing a concrete direction for improvement, thereby helping the agent learn from prior errors and perform better on the task.

Different from reflections in single-agent systems, we integrate agent profiles into the multi-agent reflection process to obtain role-specific reflections, and take a fully observable setting to assist the agent in error detection by offering interaction histories from each agent's perspective. The reflection process of the agent can be defined as:

$$y_{k,\lambda}^i = \text{Reflector}^i(p^i, [sm_{k,\lambda,T}^i]_{i=1}^N, r_{k,\lambda}), \tag{4}$$

where $k$ represents the problem, $\lambda$ indicates $\lambda$-th trial of answer to question $k$, $T$ is the length of the trajectory $\tau_{k,\lambda}$ and $r_{k,\lambda}$ is the environmental rewards. Due to the iterative updating nature of the short-term memory, $sm_{k,\lambda,T}^i$ contains the complete action information of agent $i$ in trajectory $\tau_{k,\lambda}$.

We store all previous reflections of agent $i$ in its long-term memory, which are then added as additional context of the actor model. The decision-making process in Equation 3 can be further defined as:

$$a_{k,\lambda,t} = \text{Actor}^i(p^i, lm_{k,\lambda}^i, sm_{k,\lambda,t}^i, s_{k,\lambda,t}), \tag{5}$$

where we additionally incorporate subscript $\lambda$ due to the introduction of the reflection mechanism.

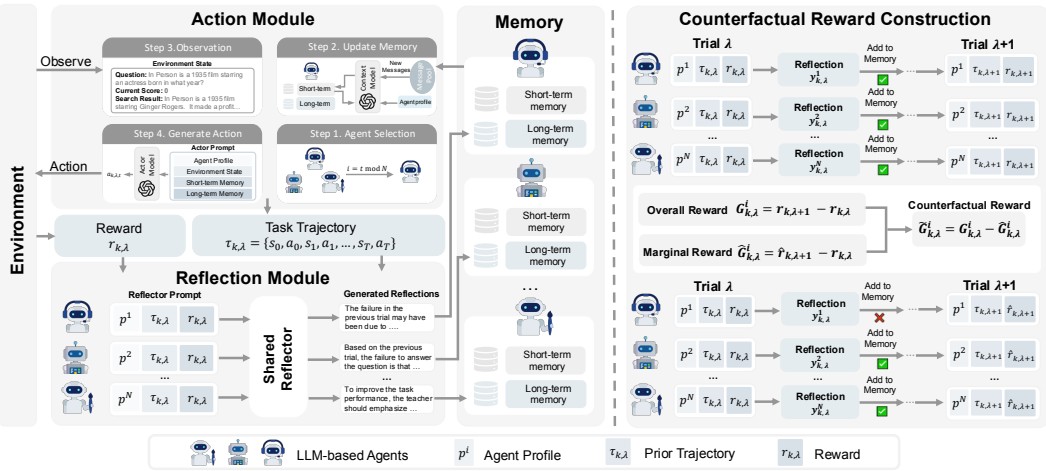

Figure 2: The overview of our proposed COPPER. The left side illustrates the multi-agent reflection framework. The system first computes the identifier $i$ of the agent to respond at the current time (Step 1). Then, agent $i$ updates its memory, including reflections of previous trials and the current trial's historical interactions (Step 2), perceives the environmental state such as the question and current task scores (Step 3), and generates the action (Step 4). After several rounds of interaction, the task trajectory and the reward score are fed into reflectors along with agent profiles to generate reflections, which are then stored in long-term memories and serve as additional context for the continuous optimization of actor prompts. On the right side, we depict the construction of counterfactual rewards, which are further employed for fine-tuning the shared reflector.

## 4.3 Optimization of the Shared Reflector

Generating useful reflective feedback with frozen LLMs in multi-agent systems proves to be challenging, since it demands a profound grasp of agent characteristics and collaborative environments. Hence, in this paper, we propose to fine-tune a shared reflector using open-source LLMs (such as Llama) with our counterfactual enhanced proximal policy optimization mechanism.

### 4.3.1 Instruction and Response Collection

In the episode $\lambda$ of problem $k$, the multi-agent system first interacts with the environment to produce a trajectory $\tau_{k,\lambda}$, after which the reward function returns a score $r_{k,\lambda}$. Agents in the system then reflect on the prior failed trajectory and generate verbal feedback to refine the corresponding actor prompt. In the process, Reflector$^i$ takes $\{p^i, [sm^i_{k,\lambda,T}]^N_{i=1}, r_{k,\lambda}\}$ as the instruction $x^i_{k,\lambda}$ and is prompted to produce a reflection response $y^i_{k,\lambda}$. Considering the homogeneity between agent reflectors as well as the training efficiency, we gather reflection data from all agents across tasks and trials to train a shared reflector. The offline training data $D$ can be defined as follows:

$$D = \{(x^i_{k,\lambda}, y^i_{k,\lambda}) | 1 \le i \le N, 1 \le \lambda \le \Lambda, 1 \le k \le K\}, \tag{6}$$

where $\Lambda$ is the maximum trial count and $K$ is the total number of problems.

### 4.3.2 Counterfactual Reward

In this paper, to alleviate the credit assignment issue, we propose the counterfactual reward to achieve agent-specific reflection ratings for multi-agent collaboration. The construction of counterfactual rewards is shown on the right side of Figure 2.

Specifically, we first calculate an overall reward of the multi-agent system $G^i_{k,\lambda}$ following Retroformer [38], i.e, $G^i_{k,\lambda} = r_{k,\lambda+1} - r_{k,\lambda}$. Then, we sequentially marginalize out a piece of reflection from agent $i$ (which means we do not add the reflection to the actor model's prompt in the subsequent trial), while keeping other agents' reflections fixed. A new reward score $\hat{r}_{k,\lambda+1}$ is then returned after an interaction trajectory, based on which we calculate a marginal reward $\hat{G}^i_{k,\lambda} = \hat{r}_{k,\lambda+1} - r_{k,\lambda}$. Finally, the counterfactual reward of a reflection pair $(x^i_{k,\lambda}, y^i_{k,\lambda})$ is calculated by subtracting the

marginal reward from the overall reward:

$$\tilde{G}^i_{k,\lambda} = G^i_{k,\lambda} - \hat{G}^i_{k,\lambda}. \tag{7}$$

Our counterfactual dataset $D_{CF}$ can be further denoted as:

$$D_{CF} = \{(x^i_{k,\lambda}, y^i_{k,\lambda}, \tilde{G}^i_{k,\lambda}) | 1 \le i \le N, 1 \le \lambda \le \Lambda, 1 \le k \le K\}. \tag{8}$$

### 4.3.3 Counterfactual Proximal Policy Optimization

Following previous works that tackle Reinforcement Learning from Human Feedback (RLHF) [21], we adopt a similar three-step approach to fine-tune the shared reflector with counterfactual rewards.

For the first step, we take the reflections with positive scores as demonstration data and train a supervised reflector $\pi^{SFT}$ with Supervised Fine-Tuning (SFT), which can be written as:

$$\mathcal{L}_{SFT}(\boldsymbol{\theta}) = -\mathbb{E}_{(x,y)\sim D_{CF}}[\sum_{k=1}^{m} \log \pi_\theta(y_k|x, y_{<k})], \tag{9}$$

where $x$ is the reflection prompt, and $y$ represents the generated reflection.

For the second step, taking construction expenses into account, instead of collecting pairwise responses for each input, we train a regression model to assess prompt and reflection pairs. We optimize the reward model $R_{CF_\phi}$ with counterfactual dataset $D_{CF}$ by minimizing the Mean Square Error (MSE) loss:

$$\mathcal{L}_{RM}(\boldsymbol{\phi}) = \mathbb{E}_{(x,y,r)\sim D_{CF}}[(R_{CF_\phi}(x,y) - r)^2]. \tag{10}$$

For the third step, we utilize the counterfactual reward model to optimize the supervised reflector via PPO. We begin by initializing $\pi^{SFT}$, which is used to produce predictions $\hat{y}$ for randomly chosen samples $x$ from the entire dataset $D_{CF}$. Subsequently, the counterfactual reward model $R_{CF_\phi}$ assigns a reward to each response. Our goal is to optimize the reflector model by maximizing the total reward, which can be accomplished by minimizing the following loss objective:

$$\mathcal{L}_{PPO}(\boldsymbol{\theta}) = -\mathbb{E}_{x\sim D_{CF}}\mathbb{E}_{y\sim\pi_\theta^{RL}(x)}[R_{CF_\phi}(x,y) - \beta \log \frac{\pi_\theta^{RL}(y|x)}{\pi^{SFT}(y|x)}]. \tag{11}$$

## 5 Experiments

### 5.1 Datasets

We choose HotPotQA [36], GSM8K [7], and Checkmate in One Move [28] to evaluate the collaborative abilities of multi-agent systems in multi-hop question answering, mathematics and chess.

**HotPotQA**   HotPotQA is a multi-hop question-answering dataset designed to evaluate models' complex reasoning ability. It contains 90,447 question-answer pairs that generally require multiple reasoning steps across documents to arrive at an answer.

**GSM8K**   GSM8K is a collection of 8.5K diverse and high-quality math word problems for grade school students. Each problem requires between 2 to 8 steps to solve, with solutions mainly involving a series of fundamental calculations with basic arithmetic operations.

**Checkmate in One Move**   Checkmate in One Move is a dataset from The Beyond the Imitation Game Benchmark (BIG-bench), featuring 3,500 games to assess language models' proficiency in playing chess using standard algebraic notation (SAN). When presented with a move sequence leading to a potential checkmate, the model is tasked with identifying the move that achieves checkmate.

### 5.2 Baselines

We compare the following baseline models to verify the effectiveness of COPPER: 1) **CoT** [30]. CoT suggests bridging the gap between question and answer by generating intermediate reasoning and is useful for simple questions without tool needs. We adopt CoT in math and chess environments

following [8] to represent the initial success rate of the system. 2) **ReAct** [37]. This is the state-of-the-art frozen language agent architecture, which mainly relies on the reasoning and planning ability of LLMs. It serves as a baseline in HotPotQA to denote how the agent performs without using environmental feedback. 3) **Reflexion** [27]. This is a classic framework to learn from environment signals and generate verbal feedback to improve task performance. We extend the method to multi-agent systems and respectively employ GPT-3.5 and LongChat as reflectors to reflect on multi-agent ReAct or CoT trajectories, without fine-tuning the reflectors. 4) **Retroformer** [38]. The paper proposes an effective method for enhancing the reflective capability of agents in single-agent systems. Here, we treat the agents in a multi-agent environment as mutually independent and fine-tune the reflector of each agent following Retroformer as a baseline.

## 5.3 Implementation Details

**Model**    We use GPT-3.5 (model: gpt-3.5-turbo) as the frozen actor models as well as the context models of agents and fine-tune LongChat (model: longchat-7b-16k) as the shared reflector. We choose gpt-2 as the regression reward model for counterfactual PPO training.

**Collaboration Settings**    We adopt a cooperative debate paradigm on GSM8K and Checkmate in One Move following [8], while on HotPotQA, in alignment with [27, 38], we design a teacher-student paradigm to enable agents to call the retrieval tool.

**Data Collection**    We randomly select 2,000 tasks to collect reflection data on HotPotQA and Checkmate in One Move, while on the GSM8K dataset, due to the higher initial success rate and fewer reflections, we randomly select 3,000 instances. We set the maximum number of trials to 5, the temperature of GPT-3.5 to 0, and the temperature of LongChat to 0.9. We use the F1 score as the reward function of HotPotQA following [38] and exact match score in other environments. Comprehensive details regarding the quantity of collected datasets can be found in Appendix B.

**Training**    We use LoRA [14] for efficient fine-tuning of the shared reflector and implement RLHF through the trl package of HuggingFace. For SFT training, we tune the epoch in $\{1, 2, 3, 4\}$, batch size in $\{64, 128, 256\}$, and learning rate in $\{1e\text{-}4, 2e\text{-}4, 3e\text{-}4, 5e\text{-}4\}$ through grid search on a validation set with 100 instances, while for counterfactual PPO, we change the search range of learning rate to $\{1e\text{-}5, 2e\text{-}5, 3e\text{-}5, 5e\text{-}5\}$. As for the reward model, we set learning rate to $5e\text{-}5$, training epoch to 3 and batch size to 16. We conduct all experiments on four NVIDIA A800-80G GPUs.

**Evaluation**    In alignment with constraints imposed by computational resources and following precedents set by earlier research [38, 27], we randomly sample 100 instances as the test set. We set the temperature of both GPT-3.5 and LongChat to 0 during the test phase to ensure reproducibility. We measure the performance of the system in exact match accuracy during the test phase.

## 5.4 Main Results

We compare the performance of COPPER against different baselines on HotPotQA, GSM8K, and Checkmate in One Move after 5 trials as main results, which are shown in Figure 3. By observing the results, we find that the results of different methods on the three datasets show roughly the same pattern: (1) Contrasted with the outcomes of multi-agent ReAct or CoT, employing the multi-agent reflection framework outlined in Section 4.2 can notably enhance the performance of multi-agent systems in specific tasks. For instance, in HotPotQA environment, the inclusion of LongChat and GPT-3.5 as reflectors leads to improvements of 15.9% and 22.7%, respectively, over the initial success rate. (2) Compared to the original LongChat and GPT-3.5, COPPER demonstrates stronger reflective abilities. The fine-tuned reflector is proficient in identifying the cause of task failure and devising personalized improvement strategies for diverse intelligent agents. Compared to the initial success rate, COPPER brought improvements of 31.8%, 18.5%, and 86.4% on the HotPotQA, GSM8K, and Checkmate in One Move datasets, respectively. (3) Compared to Retroformer, COPPER can improve the performance of multi-agent collaboration faster. We speculate that the improved performance is brought by our special designs for multi-agent settings, such as the counterfactual reward and the shared reflector.

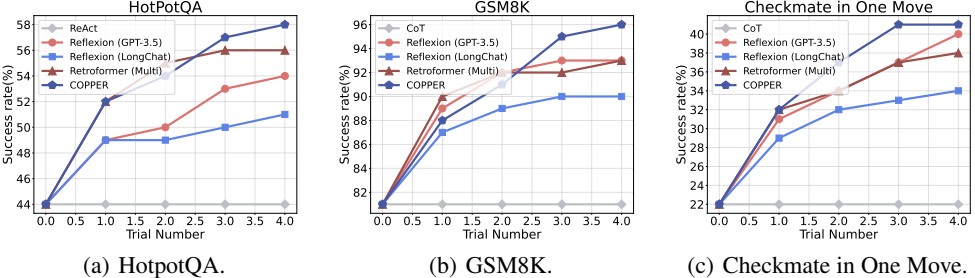

(a) HotpotQA.  (b) GSM8K.  (c) Checkmate in One Move.

Figure 3: Performance of COPPER against baselines on three datasets.

## 5.5 Ablation Study

We conduct an ablation study on three datasets to explore the effectiveness of each component of COPPER. We exclude the counterfactual reward (w/o CF) and proximal policy optimization (w/o PPO) individually and illustrate the outcome of Reflexion (LongChat) for comparison purposes (equivalent to eliminating the entire fine-tuning process). Experimental results are shown in Figure 4. From the results, we can conclude that both counterfactual reward and PPO fine-tuning are crucial for COPPER, and removing any part will lead to a decrease in performance. On the one hand, substituting counterfactual rewards with episode return difference rewards will lead to uniform rewards for all agents' reflections, meaning the contribution of reflection by each agent is equal. This could elevate the reward score for reflections that offer little assistance in enhancing collaboration performance, presenting a challenge in refining the reflector. On the other hand, fine-tuning PPO on the basis of SFT can further enhance the reflective ability of the shared reflector. This indicates that by maximizing environmental rewards, PPO can refine the model's output to better suit human preferences. For HotPotQA and GSM8K, we notice that the enhancement from COPPER during the initial two rounds is comparatively lower than solely fine-tuned with SFT. However, COPPER exhibits the highest success rate after five trials. This may be due to the fact that during the PPO training process, the reflector learns to sacrifice early performance for greater ultimate benefits.

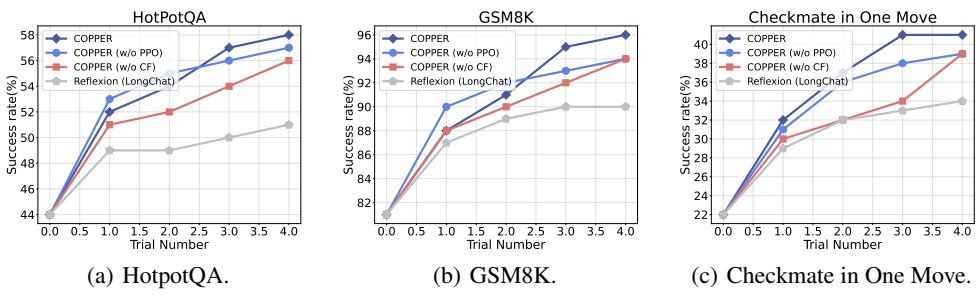

(a) HotpotQA.  (b) GSM8K.  (c) Checkmate in One Move.

Figure 4: Ablation study.

## 5.6 Generalizability of the Shared Reflector

We conduct experiments on three datasets to investigate the generalizability of COPPER, with the outcomes visualized in Figure 5. Specifically, we implement COPPER trained in multi-agent systems with GPT-3.5 actors to systems with GPT-4 (model: gpt-4-turbo) actors. We compare generalized COPPER against two baselines: one featuring GPT-4 as the reflector and the other utilizing LongChat as the reflector. We conclude that COPPER remains proficient in reflection capabilities within the systems featuring GPT-4 actors. Compared to the initial success rate, COPPER demonstrates improvements of 27.7%, 9.0%, and 53.3% in HotPotQA, GSM8K, and Checkmate in One Move respectively, and achieves comparable performance to GPT-4 reflectors after 5 trials.

## 5.7 Generality of Counterfactual Rewards

To tackle the credit assignment challenge in multi-agent systems, the paper suggests deriving scores for individual agent reflections using counterfactual rewards, which is essentially a data augmentation approach. Hence, in this section, we delve into the suitability of counterfactual rewards for LLM fine-tuning techniques beyond RLHF. Specifically, we evaluate the performance of CF SFT (employing

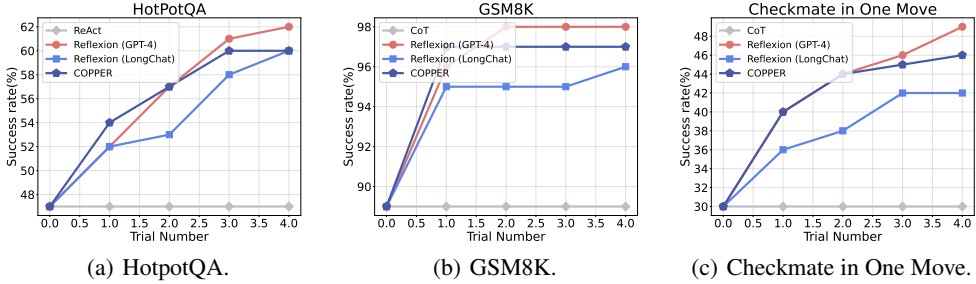

(a) HotpotQA.  (b) GSM8K.  (c) Checkmate in One Move.

Figure 5: Apply the shared reflector trained for GPT-3.5 to GPT-4.

counterfactual rewards to screen positive examples) against that of typical SFT fine-tuning (utilizing episode difference rewards to filter positive examples), as illustrated in Figure 6. Analysis of the results reveals that CF SFT outperforms regular SFT across all three scenarios. This underscores the effectiveness of counterfactual rewards in offering a more objective score based on model reflection contributions, thereby ensuring the selection of positive examples of higher quality.

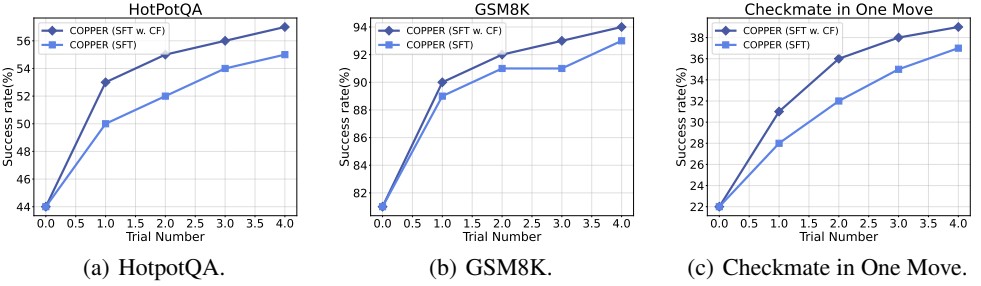

(a) HotpotQA.  (b) GSM8K.  (c) Checkmate in One Move.

Figure 6: Applying counterfactual rewards to SFT.

## 5.8 Effect of the Shared Reflector

In multi-agent systems, the quantity of reflectors will increase with the number of agents. This will lead to an excessive search space of hyper-parameters, posing challenges for practical applications. Therefore, we suggest training a shared reflector that employs carefully designed prompts to enhance the training efficiency and stability, without compromising personalized reflective abilities. In this section, we explore the effectiveness of shared reflector and present results in Figure 7. During the implementation of non-shared reflectors, given the uniformity among agents, we streamline the hyper-parameter search by aligning the hyper-parameters of each reflector. Experiments indicate that shared reflector can deliver better reflection effects, possibly because it can access more training data, leading to superior training outcomes.

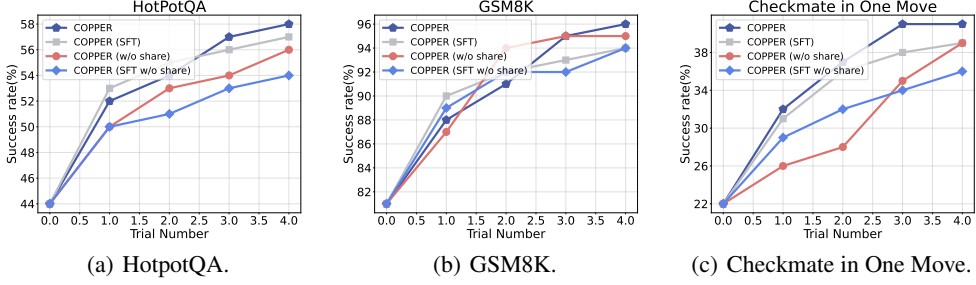

(a) HotpotQA.  (b) GSM8K.  (c) Checkmate in One Move.

Figure 7: Exploring the effectiveness of shared reflector.

## 5.9 Effectiveness of Agents' Profiles

In order to reduce training costs while generating personalized reflections in multi-agent systems, we propose to add agent profiles to the input of reflectors and train a shared reflector. In this section, we further verify the necessity of role information in multi-agent reflection scenarios. Specifically, we remove the agents' profiles from the input of the reflector, and the experimental results are shown

in Figure 8. By comparing Figure 3 and Figure 8, we can observe that when using pre-trained LMs (LongChat and GPT-3.5) to reflect, the removal of the agent profile has a greater impact on the GPT-3.5 reflector. This may be due to GPT-3.5's better contextual understanding ability. Our COPPER can further improve the model's reflection ability under no-profile setting. However, the results are slightly worse than the setting with agent profiles.

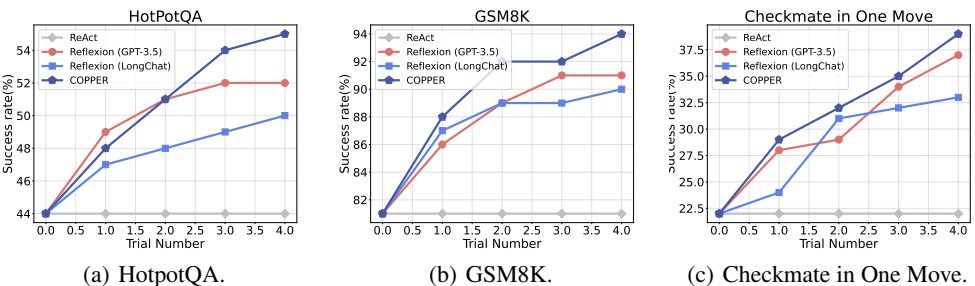

(a) HotpotQA.       (b) GSM8K.       (c) Checkmate in One Move.

Figure 8: Performance under no-profile setting.

## 5.10 Different LLMs as Base Reflectors

In this section, we replace the base reflector from LongChat with Llama-3 (model: llama-3-8b-16k) and explore the applicability of COPPER for different base models on the GSM8K dataset. Experimental results shown in Figure 9 demonstrate that our proposed COPPER has good performance across different base models. When comparing to the initial success rate, fine-tuning Llama-3 with counterfactual PPO shows a 17.3% enhancement, surpassing the performance of the GPT-3.5 reflector after 5 trials. Additionally, we include the outcome from fine-tuning Llama-3 exclusively with SFT. From the results, we find that PPO can further improve the reflective capabilities of the shared reflector.

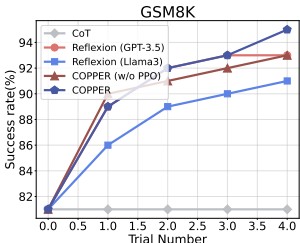

Figure 9: Fine-tuning Llama-3 as the shared reflector.

## 6 Limitations

While counterfactual rewards can mitigate the credit assignment issue in multi-agent collaboration, constructing such rewards with LLMs imposes additional data requirements. Though our proposal involves training a shared reflector and updating the reward model's loss function to MSE, investigating more efficient data collection approaches is still needed. Besides, in this study, we restrict long-term memory to a sliding window with a maximum capacity. We believe extending the agent's memory to more advanced structures such as vector embeddings presents a promising direction for development.

## 7 Conclusion

In this paper, we consider leveraging the self-reflection mechanism to improve multi-agent collaboration, and propose an elegant framework COPPER. Towards more efficient reflection, we train a shared reflector using the counterfactual PPO mechanism. The counterfactual reward can be evaluated according to the impact of each agent reflection on enhancing task performance. To enhance the training efficiency and stability, we gather reflection data across agents and train a shared reflector. Experiments on three datasets indicate that our COPPER exhibits superior reflective ability and effective generalization across various actor models.

## Acknowledgements

This work is supported in part by National Key R&D Program of China (2023YFF0905402), National Natural Science Foundation of China (No.62102420), Intelligent Social Governance Platform, Major Innovation & Planning Interdisciplinary Platform for the "DoubleFirst Class" Initiative, Renmin University of China, Public Computing Cloud, Renmin University of China, fund for building world-class universities (disciplines) of Renmin University of China, Intelligent Social Governance Platform. The work is also supported by Huawei Innovation Research Programs. We gratefully acknowledge

the support from Mindspore[2], CANN (Compute Architecture for Neural Networks) and Ascend AI Processor used for this research.

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

# A  Collaboration Settings

In HotPotQA scenario, we adopt a teacher-student collaboration paradigm, while on GSM8K and Checkmate in One Move datasets, we employ a collaborative debate setting follow [8]. We illustrate the process of the collaborations in Figure 10 and detailed introductions are as follows.

**HotPotQA**  In the HotPotQA scenario, both teacher and student agents adopt the ReAct method for action selection. Among them, student agents can call retrieval tools to search for relevant text segments provided by the HotPotQA dataset. The retrieval tool is constructed using SimCSE (model: unsup-simcse-roberta-base).

The action space of student agent includes: **(1) Search[entity]**, which invokes a local searcher to provide relevant information. **(2) Finish[answer]**, which returns the answer and finishes the task.

The action space of teacher agent includes: **(1)[Continue]**, which means the student made a good decision and should continue the process. **(2)[Rethink]**, which means the student's previous step is wrong and should consider another step.

**GSM8K**  In the GSM8K scenario, we set three debater agents to engage in two rounds of debate, and the final answer of the system is determined by voting on the last round answers. During the debate, intelligent agents utilize CoT to analyze and answer questions, or update the answer based on solutions of other agents.

**Checkmate in One Move**  In the Checkmate in One Move scenario, we adopt the same collaborative debate approach to construct multi-agent system as GSM8K. The only difference is that we employ three intelligent agents for three rounds of debate.

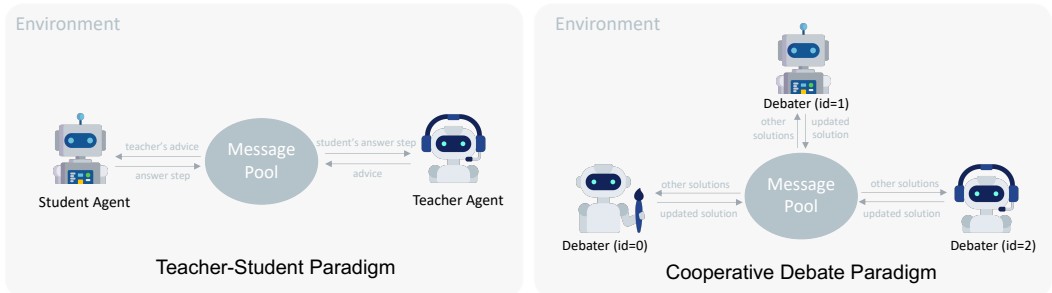

Figure 10: Multi-agent collaboration framework.

# B  Details of Training Data

We first use the original LongChat to construct counterfactual training data and select positive examples for supervised fine-tuning of the reflector model. The detailed data information generated in this stage is shown in Table 1.

Table 1: Statistics of training data generated by original LongChat. We show the total data volume on the left, the number of positive examples on the middle, and the number of negative examples on the right side.

| Dataset | Episode Difference Reward | | | | Counterfactual Reward | | | |
|---|---|---|---|---|---|---|---|---|
| | all | agent_0 | agent_1 | agent_2 | all | agent_0 | agent_1 | agent_2 |
| HotPotQA | 8714/1946/1240 | 4357/973/620 | 4357/973/620 | -/-/- | 8714/2137/1330 | 4357/1065/686 | 4357/1072/644 | -/-/- |
| GSM8K | 3147/852/0 | 1049/284/0 | 1049/284/0 | 1049/284/0 | 3147/871/417 | 1049/138/145 | 1049/128/131 | 1049/132/141 |
| Checkmate. | 17466/906/0 | 5822/302/0 | 5822/302/0 | 5822/302/0 | 17466/682/461 | 5822/234/165 | 5822/223/137 | 5822/228/159 |

Afterwards, we employ the reflector model fine-tuned using counterfactual SFT to generate training data of the PPO stage. We use the collected counterfactual reward data to train a reward model using

MSE loss and score the reflections predicted during the PPO training process. The data collected in this stage is shown in Table 2.

Table 2: Statistics of training data generated by LongChat fine-tuned with SFT. We show the total data volume on the left, the number of positive examples on the middle, and the number of negative examples on the right side.

| Dataset | Episode Difference Reward | | | | Counterfactual Reward | | | |
| --- | --- | --- | --- | --- | --- | --- | --- | --- |
| | all | agent_0 | agent_1 | agent_2 | all | agent_0 | agent_1 | agent_2 |
| HotPotQA | 8822/2218/1410 | 4411/1109/705 | 4411/1109/705 | -/-/- | 8822/1679/1985 | 4411/855/1027 | 4411/824/958 | -/-/- |
| GSM8K | 3204/789/0 | 1068/263/0 | 1068/263/0 | 1068/263/0 | 3204/384/409 | 1068/129/143 | 1068/128/132 | 1068/127/134 |
| Checkmate. | 17892/591/0 | 5964/197/0 | 5964/197/0 | 5964/197/0 | 17892/412/444 | 5964/138/147 | 5965/138/169 | 5966/136/128 |

# C    Experimental Results on ALFWorld

ALFWorld is a classical dataset designed for training and evaluating AI agents in interactive environments. To further improve our study, we additionally conduct experiments on ALFWorld. We follow the same multi-agent collaboration setting as [31] and test the model with 134 instances. The experiment results are presented in Figure 11. From the results, we can observe that COPPER achieves better reflection performance than the original LongChat and GPT-3.5. Besides, compared to the initial success rate, COPPER brings an improvement of 37.2% with 4 times of reflections.

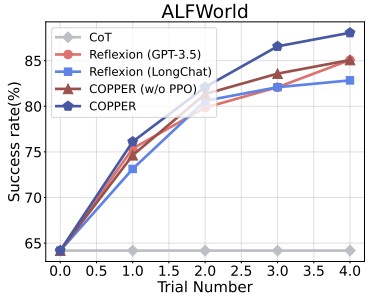

Figure 11: Experimental results on ALFWorld.

# D    Comparison between SFT and RLHF under No-profile Setting

We also provide experimental results comparison of different fine-tuning methods under no profile setting. The results are shown in Figure 12. We can find that even without agent profiles, the PPO method still provides enhancement to the fine-tuning of the shared reflector, further validating the effectiveness of our method.

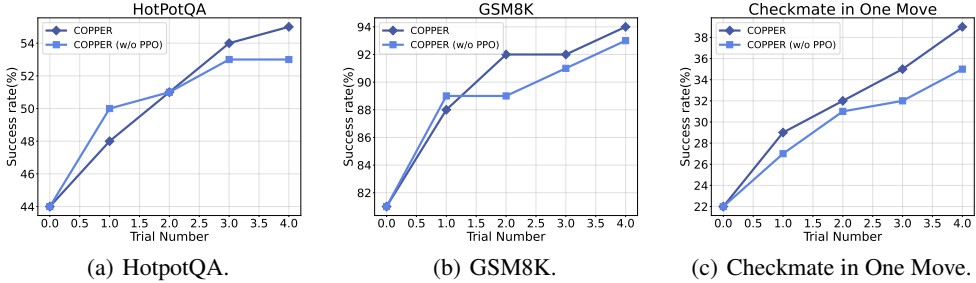

(a) HotpotQA.    (b) GSM8K.    (c) Checkmate in One Move.

Figure 12: Comparison between SFT and RLHF under no-profile setting.

# E    Experiments under Partial Information Settings

To improve our paper, we conduct more experiments to investigate the settings with partial information. In specific, we introduce two models. For the first one, we remove the information of the other agents.

For the second one, we use a proxy model to predict the information of the other agents. We present the experiment results in Figure 13 and Figure 14, respectively. We can find from the results that our COPPER still achieves better reflection performance under these partial settings.

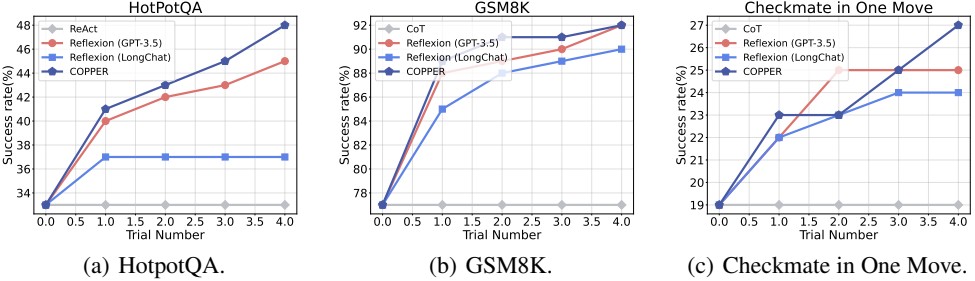

(a) HotpotQA.  (b) GSM8K.  (c) Checkmate in One Move.

Figure 13: Main results without other agents' trajectories.

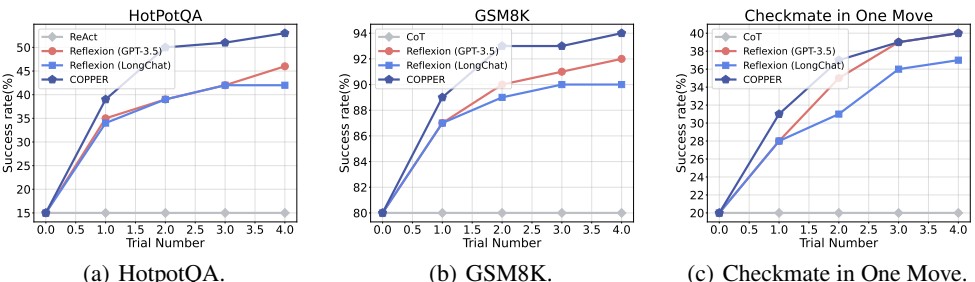

(a) HotpotQA.  (b) GSM8K.  (c) Checkmate in One Move.

Figure 14: Main results when predicting other agents' information with a proxy model.

# F  Case Study

We conduct case studies on three datasets to visually verify the effectiveness of our method, as shown in Figure 15, 16 and 17.

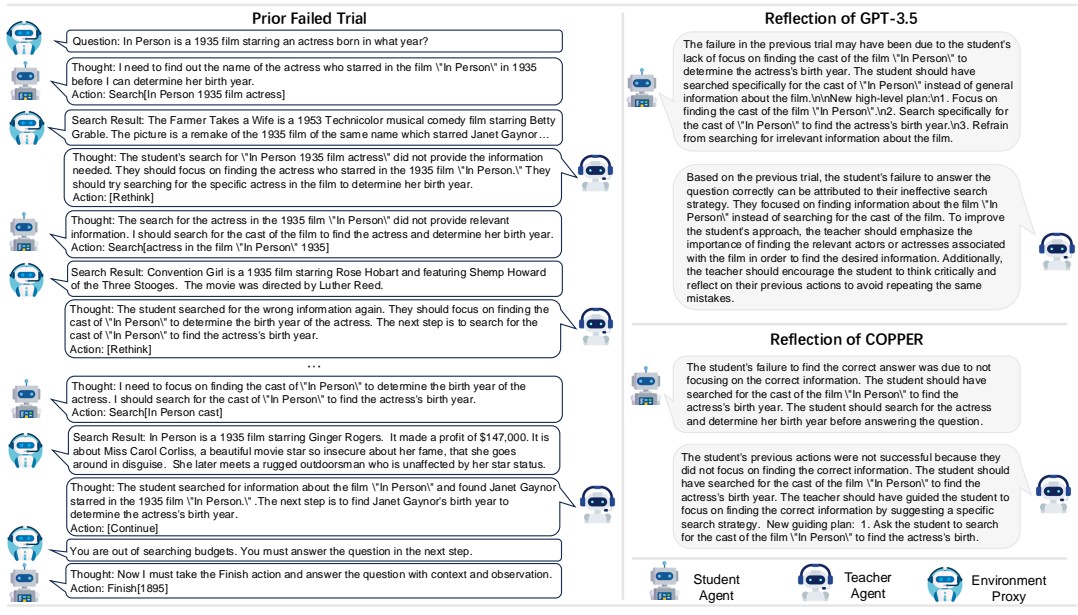

Figure 15: Case study on HotPotQA dataset.

For example, Figure 15 shows a prior failed trajectory in HotPotQA scenario, as well as the reflections generated by COPPER and GPT-3.5. During interactions with the environment, the student agent

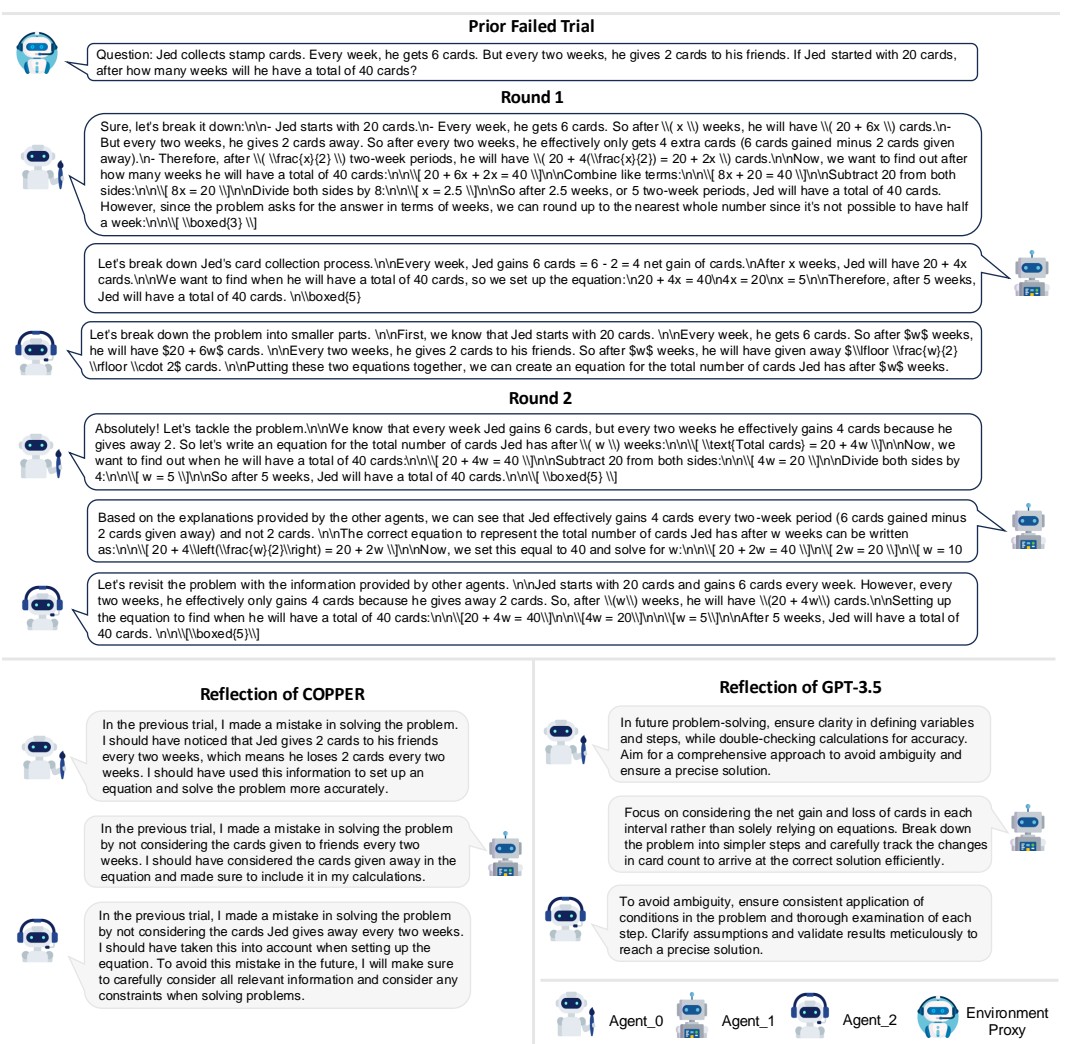

Figure 16: Case study on GSM8K dataset.

needs to constantly modify keywords of the retriever API to obtain information related to the problem. However, in the previous trajectory, the student agent underwent multiple modifications to complete the first step of inference, which results in reaching the pre-defined maximum search step limit. Therefore, the agent is forced to answer the question directly based on search result of the first reasoning step, leading to an incorrect answer. By analyzing the reflections of COPPER and GPT-3.5, we can observe that: (1) Compared to GPT-3.5, COPPER has a deeper understanding of agent roles and thus generate corresponding reflections for each agent. For example, in the reflection of the teacher agent, the reflection generated by GPT-3.5 contains lots of descriptions of problems related to the student agent, while COPPER focuses more on reflecting on the role of the teacher. (2) Compared to GPT-3.5, COPPER can conduct more effective and comprehensive root cause analysis to identify the cause of failure. In the reflection of the student agent, both GPT-3.5 and COPPER mention that "the case of In Person" should be used as the search keyword, but COPPER additionally suggests that the relevant information of the actress should be retrieved before answering the question.

As for the case studies on GSM8K (Figure 16) and Checkmate in One Move (Figure 17) datasets, we can find that in the context of the multi-agent debate, compared to GPT-3.5, COPPER can accurately identify key issues where mistakes occurred in previous trials and generate more specific reflections to guide agents in improving their responses. For example, in the case of GSM8K, COPPER identified that previous errors mainly stemmed from an incorrect analysis of equal relations and emphasized

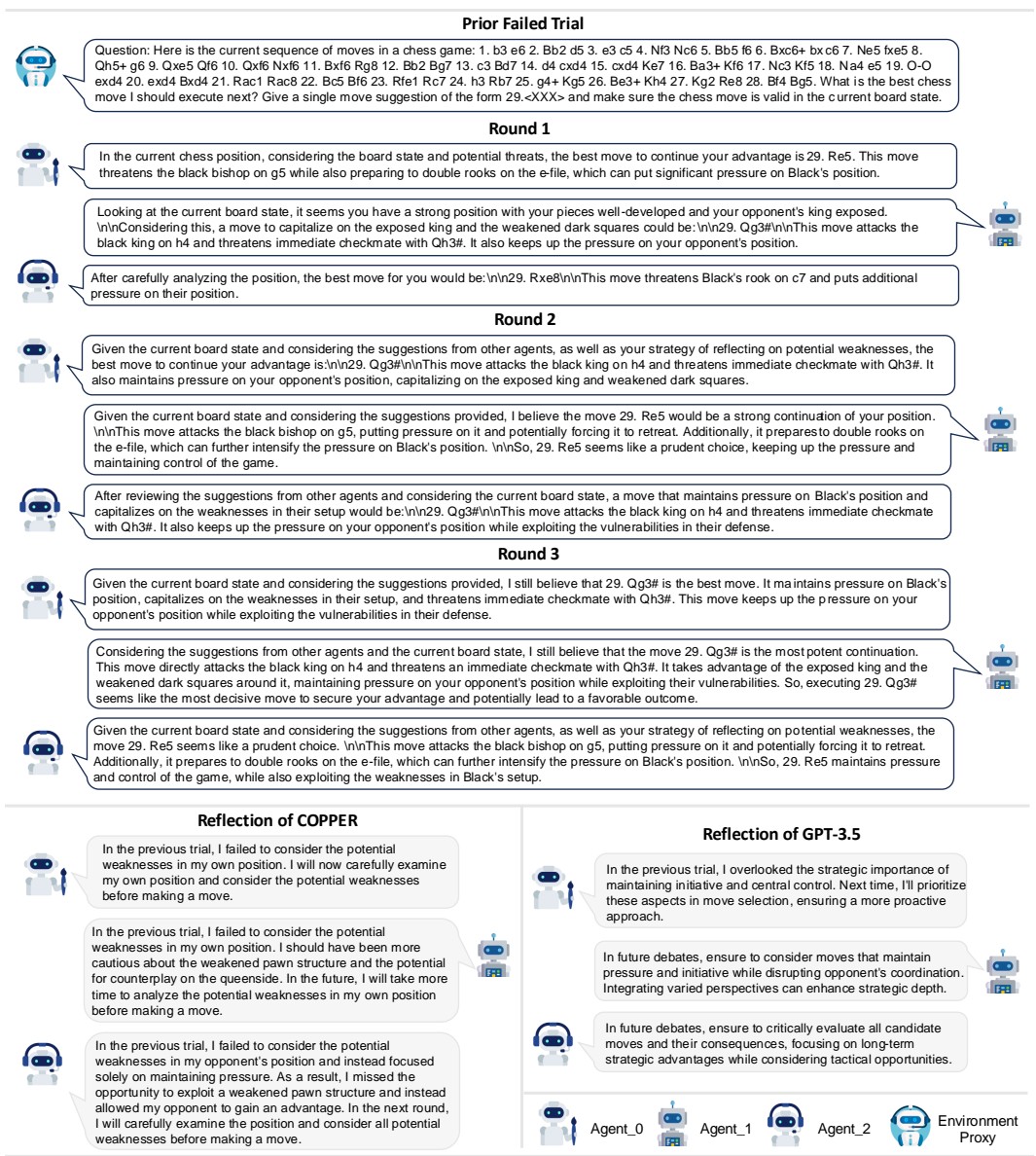

Figure 17: Case study on Checkmate in One Move dataset.

the need to pay attention to "every two weeks". In the case of Checkmate in One Move, COPPER pointed out the importance of focusing on the weakness in the own position.

# G   Ethical Consideration

In this paper, we propose using the self-reflection mechanism to enhance the collaborative ability of multi-agent systems, and explore the effectiveness of our method in three scenarios: question answering, mathematics, and chess. Experimental results show that our method can effectively enhance the ability of multi-agent systems to solve complex tasks, which helps to enhance the application of multi-agent systems in real-world scenarios, such as disaster response, intelligent transportation systems, and other scenarios. However, in the task of question-answering, the agent can call API to utilize retrieval tools, which may pose potential risks, such as tampering with the information in Wikipedia during the searching process. However, during the implementation, we limit the text content that intelligent agents can access, thus avoiding this issue.

# H Prompts

## H.1 Single-Agent Prompts

### H.1.1 HotPotQA

For single-agent setting on HotPotQA, we adopt the same few-shot examples as [27].

---

**The actor prompt for single-agent ReAct.**

Solve a question answering task with interleaving Thought, Action, Observation steps. Thought can reason about the current situation, and Action can be three types:
(1) Search[entity], which searches the exact entity on Wikipedia and returns the first paragraph if it exists. If not, it will return some similar entities to search.
(2) Finish[answer], which returns the answer and finishes the task. You may take as many steps as necessary.

Here are some examples:
{examples}
(END OF EXAMPLES)

Question: {question}
{scratchpad}

---

**The actor prompt for single-agent Reflexion.**

Solve a question answering task with interleaving Thought, Action, Observation steps. Thought can reason about the current situation, and Action can be two types:
(1) Search[entity], which searches the exact entity on Wikipedia and returns the first paragraph if it exists. If not, it will return some similar entities to search.
(2) Finish[answer], which returns the answer and finishes the task. You may take as many steps as necessary.

Here are some examples:
{examples}
(END OF EXAMPLES)

{reflections}

Question: {question}{scratchpad}

---

> **The reflector prompt for single-agent Reflexion.**
>
> You are an advanced reasoning agent that can improve based on self refection. You will be given a previous reasoning trial in which you were given access to an Docstore API environment and a question to answer. You were unsuccessful in answering the question either because you guessed the wrong answer with Finish[<answer>], or you used up your set number of reasoning steps. In a few sentences, Diagnose a possible reason for failure and devise a new, concise, high level plan that aims to mitigate the same failure. Use complete sentences.
>
> Here are some examples:
> {examples}
>
> Previous trial:
> Question: {question}
> {scratchpad}
>
> Reflection:

### H.1.2   GSM8K

We adopt a zero-shot setting on GSM8K and below are the prompts.

> **The actor prompt for single-agent CoT.**
>
> Can you solve the following math problem?
> {question}
> Explain your reasoning. Your final answer should be a single numerical number, in the form \\boxed{{answer}}, at the end of your response.

> **The actor prompt for single-agent Reflexion.**
>
> Reflections of previous trials:
> {reflections}
>
> Can you solve the following math problem?
> {question}
> Explain your reasoning. Your final answer should be a single numerical number, in the form \\boxed{{answer}}, at the end of your response.

> **The reflector prompt for single-agent Reflexion.**
>
> You are an advanced reasoning agent that can improve based on self refection. In previous trial, your task is to solve a math problem. However, you were unsuccessful in the previous trial. Now given previous interactions, you need to carefully examine the problem-solving ideas and calculation results, and form a reflection to avoid these problems in the next round. The reflection should be less than 50 words.
>
> Previous Question:
> {question}
>
> Previous interactions:
> {context}
>
> Reflection:

### H.1.3 Checkmate in One Move

We also adopt a zero-shot setting on Checkmate in One Move and below are the prompts.

---

**The actor prompt for single-agent CoT.**

Here is the current sequence of moves in a chess game:
{question}
What is the best chess move I should execute next?
Give a single move suggestion of the form {answer_step}.<XXX> and make sure the chess move is valid in the current board state.

---

**The actor prompt for single-agent Reflexion.**

Reflections of previous trials:
{reflections}

Here is the current sequence of moves in a chess game:
{question}
What is the best chess move I should execute next?
Give a single move suggestion of the form {answer_step}.<XXX> and make sure the chess move is valid in the current board state.

---

**The reflector prompt for single-agent Reflexion.**

You are an advanced reasoning agent that can improve based on self refection. In previous trial, your task is to give the best next chess move. However, you were unsuccessful in the previous trial. Now given previous interactions, you need to carefully examine the problem-solving ideas and calculation results, and form a reflection to avoid these problems in the next round. The reflection should be less than 50 words.

Previous Game:
{question}

Previous interactions:
{context}

Reflection:

## H.2 Multi-Agent Prompts

### H.2.1 HotPotQA

We follow [38] to design the prompts in this scenario.

---

**The actor prompt of the student agent when deploying ReAct.**

You are a student agent. Your task is to answer the question under the guidance of the teacher agent. You should make a reasonable plan at first.
Solve the task below with interleaving Thought, Action, Context steps. Context is the summary of historical interactions. Thought can reason about the current situation.
Your action can be two types:
(1) Search[entity], which invokes a local searcher to provide you with relevant information.
(2) Finish[answer], which returns the answer and finishes the task.

Please note: You only need to complete the thought step and output Search [Entity] in the action step, and we will return the relevant content in "Observation" for you. If you find an answer, submit it via "Finish [answer]". Identical searches will only return similar content. If the returned content has no relevant information, please actively try searching for different keywords. When submitting your answer, please try to submit the full answer if you think it is ambiguous, e.g. "movie director" is better than "director"! The answer to the question should be as accurate and concise as possible, i.e. try to answer the question with phrases instead of long sentences. Please answer yes-no question with either "yes" or "no".

Examples:
Context: Searched for Arthur's Magazine and found it was started in 1844. Teacher agreed with previous action and suggested finding the founding date of First for Women. Searched for First for Women. Question: Which magazine was started first Arthur's Magazine or First for Women?
Observation: Search Result: Search[First for Women] First for Women is a woman's magazine published by Bauer Media Group in the USA.[1] The magazine was started in 1989. Teacher's Suggestion: [Continue] Now you should answer the question.
Thought: First for Women was started in 1989. 1844 (Arthur's Magazine) < 1989 (First for Women), so Arthur's Magazine was started first.
Action: Finish[Arthur's Magazine]
END OF EXAMPLES

Context: {context}
Question: {question}
Observation: {observation}

---

You are a teacher agent, and your task is to guide the student agent to answer the questions. You should analyze whether the student's step is logically helpful, provide an analysis, and give your final action. You can also give advice on the student's future step.
Solve the task below with interleaving Thought, Action, Context steps. Context is the summary of historical interactions. Thought can reason about the current situation.
Your action can be two types:
(1) [Continue], which means the student made a good decision and should continue the process.
(2) [Rethink], which means the student's previous step is wrong and should consider another step.

Please note: You can analyze the correctness of student agent's action, summarize the useful information the student found, or provide suggestions for subsequent steps.
Your action step only have two types: [Continue] or [Rethink]. You can only take one of the above actions. Most of the time, please be an encouraging teacher, that is, unless the student is completely wrong, use [Continue] action more often.

Examples:
Context: Searched for Arthur's Magazine and found it was started in 1844. Teacher's advice is continue.
Question: Which magazine was started first Arthur's Magazine or First for Women?
Observation: Previous advice: [Continue] Finding the start time of Arthur's Magazine is helpful. Next the student should search First for Women and find its founding date. Student's action: Search[First for Women] First for Women is a woman's magazine published by Bauer Media Group in the USA.[1] The magazine was started in 1989.
Thought: The founding date of First for Women is helpful. The student should take the Finish step next.
Action: [Continue]
END OF EXAMPLES

Context: {context}
Question: {question}
Observation: {observation}

You are a student agent. Your task is to answer the question under the guidance of the teacher agent. You should make a reasonable plan at first.

Solve the task below with interleaving Thought, Action, Context steps. Context is the summary of historical interactions. Thought can reason about the current situation.

Your action can be two types:

(1) Search[entity], which invokes a local searcher to provide you with relevant information.

(2) Finish[answer], which returns the answer and finishes the task.

Please note: You only need to complete the thought step and output Search [Entity] in the action step, and we will return the relevant content in "Observation" for you. If you find an answer, submit it via "Finish [answer]". Identical searches will only return similar content. If the returned content has no relevant information, please actively try searching for different keywords. When submitting your answer, please try to submit the full answer if you think it is ambiguous, e.g. "movie director" is better than "director"! The answer to the question should be as accurate and concise as possible, i.e. try to answer the question with phrases instead of long sentences. Please answer yes-no question with either "yes" or "no".

Reflections of previous trials:
{reflections}

Examples:
Context: Searched for Arthur's Magazine and found it was started in 1844. Teacher agreed with previous action and suggested finding the founding date of First for Women. Searched for First for Women. Question: Which magazine was started first Arthur's Magazine or First for Women?
Observation: Search Result: Search[First for Women] First for Women is a woman's magazine published by Bauer Media Group in the USA.[1] The magazine was started in 1989. Teacher's Suggestion: [Continue] Now you should answer the question.
Thought: First for Women was started in 1989. 1844 (Arthur's Magazine) < 1989 (First for Women), so Arthur's Magazine was started first.
Action: Finish[Arthur's Magazine]
END OF EXAMPLES

Context: {context}
Question: {question}
Observation: {observation}

The actor prompts of the teacher agent when deploying Reflexion.

You are a teacher agent, and your task is to guide the student agent to answer the questions. You should analyze whether the student's step is logically helpful, provide an analysis, and give your final action. You can also give advice on the student's future step.
Solve the task below with interleaving Thought, Action, Context steps. Context is the summary of historical interactions. Thought can reason about the current situation.
Your action can be two types:
(1) [Continue], which means the student made a good decision and should continue the process.
(2) [Rethink], which means the student's previous step is wrong and should consider another step.

Please note: You can analyze the correctness of student agent's action, summarize the useful information the student found, or provide suggestions for subsequent steps.
Your action step only have two types: [Continue] or [Rethink]. You can only take one of the above actions. Most of the time, please be an encouraging teacher, that is, unless the student is completely wrong, use [Continue] action more often.

Reflections of previous trials:
{reflections}

Examples:
Context: Searched for Arthur's Magazine and found it was started in 1844. Teacher's advice is continue.
Question: Which magazine was started first Arthur's Magazine or First for Women?
Observation: Previous advice: [Continue] Finding the start time of Arthur's Magazine is helpful. Next the student should search First for Women and find its founding date. Student's action: Search[First for Women] First for Women is a woman's magazine published by Bauer Media Group in the USA.[1] The magazine was started in 1989.
Thought: The founding date of First for Women is helpful. The student should take the Finish step next.
Action: [Continue]
END OF EXAMPLES

Context: {context}
Question: {question}
Observation: {observation}

**The reflector prompt of the student agent when deploying Reflexion.**

You are an advanced reasoning agent that can improve based on self refection. In previous trial, you are a student agent, and your task is to answer the question under the guidance of the teacher agent. The teacher agent provides guidance on your step and explains the reasons. You were unsuccessful in answering the question either because you guessed the wrong answer with Finish[answer], or you used up the set number of reasoning steps. Now given previous interactions from student's and teacher's perspective, you should diagnose a possible reason for failure and devise a new, concise, high level plan that aims to mitigate the same failure in a few sentences.
Please note: If you believe that the previous searching and collaboration process as well as the answer were correct, please try answering the question in a different way, e.g. try to provide more concise answers, or using the same words as the question itself.

Previous trial:
Question: {question}

Interaction:
Student: {student_context}
Teacher: {teacher_context}

Reflection:

---

**The reflector prompt of the teacher agent when deploying Reflexion.**

You are an advanced reasoning agent that can improve based on self refection. In previous trial, you are a teacher agent, and your task is to guide the student agent to answer the questions. The student is unsuccessful in answering the question either because he guessed the wrong answer, or he used up the set number of reasoning steps. Now given previous interactions from student's and teacher's perspective, you should diagnose a possible reason for failure and devise a new, concise, high level guiding plan.
Please note: If you believe that the previous searching and collaboration process as well as the answer were correct, please try answering the question in a different way, e.g. try to provide more concise answers, or using the same words as the question itself.

Previous trial:
Question: {question}

Interaction:
Student: {student_context}
Teacher: {teacher_context}

Reflection:

**The prompt of the context model of the student agent.**

You are a student agent. Your task is to answer the question under the guidance of the teacher agent. Now you are provided with a previous summary, as well as new messages that were not included in the original summary. Your summary should encapsulate the main points of the new messages and integrate them into the existing summary to create a comprehensive recap. Highlight the key issues discussed, decisions made, and any actions assigned. Record the helpful factual information given by the search engine.
Please ensure that the final summary does not exceed {char_limit} characters.

Examples:
Question: Which magazine was started first Arthur's Magazine or First for Women?
Previous Summary: Searched for Arthur's Magazine.
New Observation: Search Result: Search[Arthur's Magazine] Arthur's Magazine (1844-1846) was an American literary periodical published in Philadelphia in the 19th century. Teacher's Suggestion: [Continue] Finding the start time of Arthur's Magazine is helpful. Next the student should search First for Women and find its founding date.
New Thought: Arthur's Magazine was started in 1844. I need to search First for Women next.
New Action: Search[First for Women]
Summary: Searched for Arthur's Magazine and found it was started in 1844. Teacher agreed with previous action and suggested finding the founding date of First for Women. Searched for First for Women.
END OF EXAMPLES

Question: {question}
Previous Summary: {context}
New Observation: {observation}
New Thought: {thought}
New Action: {action}
Summary:

> **The prompt of the context model of the teacher agent.**
>
> You are a teacher agent, and your task is to guide the student agent to answer the questions. Now you are provided with a previous summary, as well as new messages that were not included in the original summary. Your summary should encapsulate the main points of the new messages and integrate them into the existing summary to create a comprehensive recap. Highlight the key issues discussed, decisions made, and any actions assigned. Record the helpful factual information given by the search engine.
> Please ensure that the final summary does not exceed {char_limit} characters.
>
> Examples:
> Question: Which magazine was started first Arthur's Magazine or First for Women?
> Previous Summary: The student searched for Arthur's Magazine and found it was started in 1844. The action is helpful and the next step is finding the founding date of First for Women.
> New Observation: Previous advice: [Continue] Finding the start time of Arthur's Magazine is helpful. Next the student should search First for Women and find its founding date. Student's action: Search[First for Women] First for Women is a woman's magazine published by Bauer Media Group in the USA.[1] The magazine was started in 1989.
> New Thought: The founding date of First for Women is helpful. The student should take the Finish step next.
> New Action: [Continue]
> Summary: The student searched Arthur's Magazine and found it was started in 1844. The student then searched First for women and found it was started in 1989. Previous actions are helpful. The student should give the answer in the next step.
> END OF EXAMPLES
>
> Question: {question}
> Previous Summary: {context}
> New Observation: {observation}
> New Thought: {thought}
> New Action: {action}
> Summary:

## H.2.2 GSM8K

We follow [8] to design the prompts of debaters in this scenario.

> **The actor prompt for the agent to generate initial answers when deploying CoT.**
>
> Can you solve the following math problem?
> {question}
> Explain your reasoning. Your final answer should be a single numerical number, in the form \\boxed{{answer}}, at the end of your response.

> **The actor prompt for the agent to generate updated answers when deploying CoT.**
>
> These are the solutions to the problem from other agents:
> {solutions}
> Using the solutions from other agents as additional information, can you provide your answer to the math problem? The original math problem is {question}.
> Your final answer should be a single numerical number, in the form \\boxed{{answer}}, at the end of your response.

> **The actor prompt for the agent to generate initial answers when deploying Relfexion.**
>
> Reflections of previous trials:
> {reflections}
>
> Can you solve the following math problem?
> {question}
> Explain your reasoning. Your final answer should be a single numerical number, in the form \\boxed{{answer}}, at the end of your response.

> **The actor prompt for the agent to generate updated answers when deploying Reflexion.**
>
> Reflections of previous trials:
> {reflections}
>
> These are the solutions to the problem from other agents:
> {solutions}
> Using the solutions from other agents as additional information, can you provide your answer to the math problem? The original math problem is {question}.
> Your final answer should be a single numerical number, in the form \\boxed{{answer}}, at the end of your response.

> **The reflector prompt of each debater agent.**
>
> You are an advanced reasoning agent that can improve based on self refection. In previous trial, you are {role} and you are supposed to solve a math problem through debating with other agents. However, you were unsuccessful in the previous trial. Now given previous interactions, you need to carefully examine the problem-solving ideas and calculation results, and form a reflection to avoid these problems in the next round. The reflection should be less than 50 words.
>
> Previous Question:
> {question}
>
> Previous interactions:
> {context}
>
> Reflection:

> **The prompt of the context model of each agent.**
>
> Please summarize the following process in concise language, including the opinions of all agents.
>
> Please note: Please summarize the viewpoints of each agent and retain the role of the agent in the summary, such as agent_0, agent_1, agent_2.
>
> {scratchpad}
>
> Summary:

### H.2.3 Checkmate in One Move

We follow [8] to design the prompts of debaters in this scenario.

**The actor prompt for the agent to generate initial answers when deploying CoT.**

Here is the current sequence of moves in a chess game:
{question}
What is the best chess move I should execute next?
Give a single move suggestion of the form {answer_step}.<XXX> and make sure the chess move is valid in the current board state.

**The actor prompt for the agent to generate updated answers when deploying CoT.**

Here are other chess move suggestions from other agents:
{solutions}
Using the chess suggestions from other agents as additional advice, can you give me your updated thoughts on the best next chess move I should play given the chess sequence?
The current sequence of moves in a chess game is: {self.question}
Give a single move suggestion of the form {answer_step}.<XXX> and make sure the chess move is valid in the current board state.

**The actor prompt for the agent to generate initial answers when deploying Reflexion.**

Reflections of previous trials:
{reflections}

Here is the current sequence of moves in a chess game:
{question}
What is the best chess move I should execute next?
Give a single move suggestion of the form {answer_step}.<XXX> and make sure the chess move is valid in the current board state.

**The actor prompt for the agent to generate updated answers when deploying Reflexion.**

Reflections of previous trials:
{reflections}

Here are other chess move suggestions from other agents:
{solutions}
Using the chess suggestions from other agents as additional advice, can you give me your updated thoughts on the best next chess move I should play given the chess sequence?
The current sequence of moves in a chess game is: {self.question}
Give a single move suggestion of the form {answer_step}.<XXX> and make sure the chess move is valid in the current board state.

The reflector prompt of each debater agent.

You are an advanced reasoning agent that can improve based on self refection. In previous trial, you are {role} you are supposed to give the best next chess move through debating with other agents. However, you were unsuccessful in the previous trial. Now given previous interactions, you need to carefully examine the problem-solving ideas and calculation results, and form a reflection to avoid these problems in the next round. The reflection should be less than 50 words.

Previous Game:
{question}

Previous interactions:
{context}

Reflection:

The prompt of the context model of each agent.

Please summarize the following process in concise language, including the opinions of all agents.

Please note: Please summarize the viewpoints of each agent and retain the role of the agent in the summary, such as agent_0, agent_1, agent_2.

{scratchpad}

Summary:

