# OpenReview forum: "Reflective Multi-Agent Collaboration based on Large Language Models"
_NeurIPS.cc/2024/Conference — NeurIPS 2024 poster_

### Official Review · Reviewer_tdnV · 2024-06-13

**Soundness:** 2
**Presentation:** 3
**Contribution:** 2
**Rating:** 4
**Confidence:** 4

**Summary:**

The paper introduces COPPER, a novel framework designed to enhance collaboration in multi-agent systems using a learnable self-reflection mechanism. COPPER utilizes a shared reflector fine-tuned to adjust actor model prompts via a counterfactual PPO mechanism. This approach includes counterfactual rewards to address the credit assignment problem and enables the reflector to customize reflections based on agent roles, optimizing computational resources and training stability. The framework's efficacy is validated through experiments in multi-hop question answering, mathematics, and chess, demonstrating improved reflection capabilities and generalization across various actor models.

**Strengths:**

This paper is clearly written and explores a new setting — multiagent reflection. It also show improved performances on all three tasks. Using counterfactual rewards to perform PPO training sounds straightforward.

**Weaknesses:**

- My main concern is this paper involves a combination of various components and I could not clearly infer from the paper which part is most important. This makes the improvement for each part look marginal. Generally, this paper proposes a novel training method to enhance reflection, as well as use reflection-based multi-agent discussion to improve agent reasoning. I believe the method could be directly applicable to single agent scenario as reward for each agent is updated independently. Could you perform ablation in terms of single-agent?

- The test scenario focuses on single-step tasks, can this framework be applied to multi-step agent tasks like AlfWorld?

- How is the performance of COPPER compared to shared parameter & loss training for all LLMs?

**Questions:**

See Weakness.

**Limitations:**

Yes.

---

> ### Author Rebuttal · Authors · 2024-08-07
>
> To Reviewer tdnV:
>
> Thanks for your comments. We will try to alleviate your concerns one by one in the following.
>
> **Q1: My main concern is this paper involves a combination of various components and I could not clearly infer from the paper which part is most important. This makes the improvement for each part look marginal. Generally, this paper proposes a novel training method to enhance reflection, as well as use reflection-based multi-agent discussion to improve agent reasoning. I believe the method could be directly applicable to single agent scenario as reward for each agent is updated independently. Could you perform ablation in terms of single-agent?**
>
>
> Thanks for this comment. In our paper, we propose the problem of multi-agent reflection, which, to the best of our knowledge, is the first time in the area of LLM-based agents. For this problem, we have actually designed several tailored techniques to solve its special challenges, for example, designing the counterfactual rewards to obtain the real effect of different agent reflections, and so on.
>
> In our setting, the reward is determined by the trajectory composed of all the agents' actions, and the reflector also needs to consider all the agent actions to reflect. As a result, one agent's action may influence the final reward and reflection, and further impact the other agents' updates.
>
> To follow your suggestion, we use the episode difference reward as the reward for all the agents, and update the agent independently. We refer to the method as Multi-Retroformer and present the experiment results in Figure 2 of the PDF.
>
>
>
> **Q2: The test scenario focuses on single-step tasks, can this framework be applied to multi-step agent tasks like AlfWorld?**
>
> Thanks for the question. Actually, HotPotQA is a multi-step agent task, where agents can retrieve relevant knowledge by invoking search engines multiple times to ultimately answer questions. Each agent in the environment follows the ReAct framework and generates actions sequentially. The main experiment results on HotPotQA can be found in Section 5.4.
>
> To further improve our study, we additionally conduct experiments on ALFWorld. We follow the same multi-agent collaboration setting as [1] and test the model with 134 instances. The experiment results are presented in Figure 4 of the PDF.
>
> From the results, we can observe that COPPER achieves better reflection performance than the original LongChat and GPT-3.5. Besides, compared to the initial success rate, COPPER brings an improvement of 37.2% with 4 times of reflections.
>
>
>
> **Q3: How is the performance of COPPER compared to shared parameter & loss training for all LLMs?**
>
> Actually, we have compared the shared and non-shared versions of our reflector in Section 5.8. To further improve this part, we have added more experiments on the other datasets, which are shown in Figure 3 of the PDF.
>
> The experiment results indicate that in multi-agent collaboration scenarios, training a shared reflector enables higher-quality reflections, both when fine-tuned with SFT and RLHF techniques.
>
>
>
> **References:**
>
> [1] Qingyun Wu, Gagan Bansal, Jieyu Zhang, Yiran Wu, Shaokun Zhang, Erkang Zhu, Beibin Li, Li Jiang, Xiaoyun Zhang, and Chi Wang. Autogen: Enabling next-gen LLM applications via multi-agent conversation framework. *CoRR*, abs/2308.08155, 2023.

---

> ### Author Response · Authors · 2024-08-12
>
> Dear reviewer tdnV,
>
> Thanks so much for your contrastive comments, which can definitely improve our paper.
>
> We believe most of your concerns are about the experiments. To alleviate your concerns, we have added a large number of experiments to make our claims more convincing (see the one-page pdf).
>
> The discussion ddl is fast approaching, if you have further questions, we are very happy to discuss them.

---

> ### Author Response · Authors · 2024-08-13
>
> Dear reviewer tdnV,
>
> We deeply appreciate all the insightful comments you have posted, as they have greatly enhanced our paper.  To follow your advice, **for each point in the Weaknesses, we have added extensive experiments (see the one-page pdf)**.
>
> Since the rebuttal deadline is approaching rapidly, we would like to kindly inquire if we have adequately addressed your concerns. If there are more remaining issues, we would appreciate the chance to address them and work towards achieving a positive score.
>
> Really hope our efforts can be considered and alleviate your concerns.
>
>
> Thanks

---

> > ### Comment · Reviewer_tdnV · 2024-08-14
> > **Thanks for the rebuttal**
> >
> > Thanks for the rebuttal. After reading the rebuttal, I am inclined to keep my original score.

---

> > > ### Author Response · Authors · 2024-08-14
> > >
> > > Thanks so much for your feedback.
> > >
> > > We take a lot of time  (more than five days) and money (e.g., collecting data using GPT, renting servers to seed up the experiments) to conduct extensive experiments for each of your concerns in the Weaknesses (see the one-page pdf).
> > >
> > > We would like to kindly ask if these experiments have alleviated your concerns. If not, we would appreciate the chance to continue working towards a positive score.

---

### Official Review · Reviewer_ogh3 · 2024-07-13

**Soundness:** 3
**Presentation:** 3
**Contribution:** 2
**Rating:** 6
**Confidence:** 3

**Summary:**

The paper proposes a multi-agent reflection framework COPPER to solve reasoning tasks on several datasets such as HotPotQA, GSM8K, and Checkmate in One Move. The two main contributions are:
1. designing counterfactual rewards to alleviate the credit assignment problem;
2. training a shared reflector to personalize the reflection for each agent.

**Strengths:**

1. Novelty: The paper introduces counterfactual rewards from RL to LLM agents, to deal with the credit assignment problem in multi-agent cooperation.
2. Soundness: The authors conducted extensive experiments to thoroughly analyze the proposed mechanism.

**Weaknesses:**

1. The motivation of the shared reflector may not align with reality. Embodied scenarios do not allow complete information sharing with a central reflector.
2. The computation of counterfactual rewards can be very high. Every agent demands two times of simulation to calculate the rewards, and the computational costs could be much higher when the number of agents increases.
3. The claims of personalized reflection may not be completely conducted. For the Cooperative Debate Paradigm, there are no roles for the debaters.

**Questions:**

1. How to determine the number of agents for each task?
2. Can you present the computational cost? Including training and inference stages.
3. Can you provide more case studies, especially for the other two datasets?
4. Does the shared reflector take all the agents' trajectories together to reflect?

**Limitations:**

The efficiency of data collection and the length of reflection memory may limit the application of the method.

---

> ### Author Rebuttal · Authors · 2024-08-07
>
> To Reviewer ogh3:
>
> Thanks for your comments. In the following, we try to alleviate your concerns one by one.
>
> **Q1: The motivation of the shared reflector may not align with reality. Embodied scenarios do not allow complete information sharing with a central reflector.**
>
> Thanks for this comment. As an initial study for multi-agent reflection, we focus on the settings where the agents' information can be fully observed, which we believe indeed simplifies the real scenarios.
>
> However, we believe our study is also meaningful since it formally proposes the direction of considering agent reflection under multi-agent settings. Based on our study, one can easily extend to the settings where the information between agents is not fully observable.
>
> To improve our paper, we conduct more experiments to investigate the settings with partial information. In specific, we introduce two models. For the first one, we remove the information of the other agents. For the second one, we use a proxy model to predict the information of the other agents. We present the experiment results in Figure 6 and Figure 7 of the PDF, respectively. We can find from the results that our COPPER still achieves better reflection performance under these partial settings.
>
>
>
> **Q2: The computation of counterfactual rewards can be very high. Every agent demands two times of simulation to calculate the rewards, and the computational costs could be much higher when the number of agents increases. Can you present the computational cost? Including training and inference stages.**
>
> Following your suggestion, we have incorporated the computational costs of the training and inference stages.
>
> In the training stage, we first collect training reflection data and then train the reflector model via RLHF offline. So although constructing counterfactual rewards requires multiple times of simulation, the process does not incur additional computational cost. For example, when training reflector on HotPotQA dataset with one NVIDIA A800-80G, both SFT and reward model training stages can be finished in about 1 hour, while the PPO training requires about 4 hours (For reference only. Training time may vary with different hyperparameters).
>
> In the inference stage, we load the fine-tuned reflector model on GPU and call GPT API as the actor model. The duration of the stage mainly depends on the speed of calling GPT.
>
>
>
> **Q3: The claims of personalized reflection may not be completely conducted. For the Cooperative Debate Paradigm, there are no roles for the debaters.**
>
> Thanks for this comment. We believe personalization should be considered as a general term, any differences between the agents can be actually regarded as "personalization" and written into the profile. For the Cooperative Debate Paradigm, the stances, the goal of each agent, and the knowledge owned by different agents can be regarded as personalized.
>
>
>
> **Q4: How to determine the number of agents for each task?**
>
> For the HotPotQA dataset, we design a teacher-student collaboration framework, in which an intuitive number of agents is 2 (with one student agent and one teacher agent). In GSM8K and Checkmate in One Move datasets, we follow [1] to set the number of agents and debating rounds. The paper explores the best number setting in each scenario through empirical experiments.
>
>
>
> **Q5: Can you provide more case studies, especially for the other two datasets?**
>
> Following your suggestion, we have added more case studies in the PDF（Figure 8 and Figure 9）, which will be incorporated into the final version.
>
> From the cases, we can find that in the context of the multi-agent debate, compared to GPT-3.5, COPPER can accurately identify key issues where mistakes occurred in previous trials and generate more specific reflections to guide agents in improving their responses. For example, in the case of GSM8K, COPPER identified that previous errors mainly stemmed from an incorrect analysis of equal relations and emphasized the need to pay attention to "every two weeks". In the case of Checkmate in One Move, COPPER pointed out the importance of focusing on the weakness in the own position.
>
>
>
> **Q6: Does the shared reflector take all the agents' trajectories together to reflect?**
>
> Yes, in our model, the reflector takes all the agents' trajectories to reflect. To further improve our study, we also conduct more experiments on the settings where the agent only takes its own trajectories to reflect, see the answers to Q1 for more details.
>
>
>
> **References:**
>
> [1] Yilun Du, Shuang Li, Antonio Torralba, Joshua B. Tenenbaum, and Igor Mordatch. Improving factuality and reasoning in language models through multiagent debate. *CoRR*, abs/2305.14325, 2023.

---

> ### Author Response · Authors · 2024-08-12
>
> Dear reviewer ogh3,
>
> Thanks again for your detailed comments, which can definitely improve our paper.
>
> In our rebuttal, we have tried our best to alleviate your concerns and added many experiments inspired by your constructive suggestions.
>
> The discussion ddl is fast approaching, if you have further questions, we are very happy to discuss them.

---

> > ### Comment · Reviewer_ogh3 · 2024-08-12
> >
> > Thank you for the detailed response. I acknowledge the efforts to clarify the key concepts and add new results. However, regarding W2 and Q2, although the data is collected offline, the collection process itself requires more API calls due to the construction of the counterfactual reward. Therefore, the tokens used should be included as part of the cost analysis. I will raise my score to 6.

---

> ### Author Response · Authors · 2024-08-13
>
> Thanks so much for your feedback.
>
> In the following, we show the token costs of building our datasets to make our training process more clear.
>
> In HotPotQA dataset, each trajectory of the multi-agent system comprises approximately 28,672 input tokens and 3,584 output tokens, costing 0.01925 dollars. Constructing training data with original LongChat for CF SFT training costs 290.12 dollars in total, and the training data generated by LongChat fine-tuned with SFT for PPO training costs 293.24 dollars.
>
> In GSM8K dataset, a single trajectory consists of around 5,376 input tokens and 1,536 output tokens, with a cost of 0.004875 dollars. The total cost for creating training data with the original LongChat for CF SFT training is 35.08 dollars, and the training data constructed by LongChat fine-tuned with SFT for PPO training costs 35.45 dollars.
>
> In Checkmate in One Move dataset, each trajectory includes about 11,520 input tokens and 2,304 output tokens, which requires 0.009 dollars. Total expenses for generating training data with the original LongChat for CF SFT training amount to 227.59 dollars, while the cost for data constructed by LongChat fine-tuned with SFT for PPO training is 232.70 dollars.
>
> In the final version, we will definitely incorporate the above cost analysis to make our paper more clear.

---

### Official Review · Reviewer_gdf9 · 2024-07-13

**Soundness:** 3
**Presentation:** 3
**Contribution:** 3
**Rating:** 7
**Confidence:** 3

**Summary:**

This paper proposes COPPER to enhance the collaboration ability of multi-agent systems through a learnable self-reflection mechanism. It involves reflections from different agent-specific profiles. The contribution of each agent-specific reflector is measured based on their marginal reward. This reflector is shared among agents and generates personalized reflections according to agents' roles. Experimental results on several datasets demonstrate its effectiveness.

**Strengths:**

1. This paper explores the reflection on multi-agent collaboration. Previous work on reflection mainly focuses on a single LLM, ignoring the complex environment and interaction in the multi-agent system.
2. The introduction of the counterfactual reward in PPO training assigns the reward to rate each agent's reflection, helping the credit assignment problem.
3. The comprehensive analysis of the counterfactual reward, the shared reflector, and different LLMs for reflectors provide a deep insight into the proposed method.

**Weaknesses:**

1. Including the Retroformer under the multi-agent setting as one of the baselines would be better.
2. When the environment provides a sparse reward such as the credit for the reflection of different agents may become very similar. For example, the removal of all reflections may result in a counterfactual reward of 0 because both trials fail. Then the counterfactual reward may degrade to the episode reward in Retroformer.
3. With the complex PPO training, COPPER's performance is not very impressive, especially when the trial is small (in GSM8k and Checkmate in One Move  of Figure 4 and Figure 8)

**Questions:**

* The left part of Figure 2 is a little confusing due to the position of Step 1 to 4. The execution order is unclear. The text in Figure 2 is too small to be seen, especially the right part.
* Will there be any negative counterfactual reward? For example, the removal of a specific reflection will improve the performance.
* What is the impact of the agent profile on the reflector? Will a personalized reflector be better than a general reflector?

**Limitations:**

The authors discuss their potential limitations in their paper

---

> ### Author Rebuttal · Authors · 2024-08-07
>
> To Reviewer gdf9:
>
> Thanks so much for your positive comments on our manuscript. In the following, we try to alleviate your concerns in detail (we combine all the questions in the weaknesses and questions).
>
> **Q1: Including the Retroformer under the multi-agent setting as one of the baselines would be better.**
>
> Following your suggestion, we have implemented Retroformer under multi-agent setting, and compared it with our model. The experiment results are presented in Figure 2 of the PDF.
>
> From the results, we can see that our model can outperform Multi-Retroformer on all the datasets. We speculate that the improved performance is brought by our special designs for multi-agent settings, such as the counterfactual reward and so on.
>
>
> **Q2: When the environment provides a sparse reward such as the credit for the reflection of different agents may become very similar. For example, the removal of all reflections may result in a counterfactual reward of 0 because both trials fail. Then the counterfactual reward may degrade to the episode reward in Retroformer.**
>
> To begin with, the counterfactual reward aims to evaluate "the effectiveness of conducting reflection on each agent". For agent $i$, we compare the final rewards when agent $i$ uses and does not use reflections. The reward change is regarded as the real effect of the reflector on agent $i$. Basically, we decompose the overall reward improvement into sub-rewards, which are more tailored to different agent reflectors.
>
> According to the above explanations on the counterfactual reward, we can not remove all the reflections, for each agent, we only remove its corresponding reflection to obtain the counterfactual reward, while all the other reflections are still valid. This corresponds to the basic meaning of counterfactual effect: "What is the effect of one variable when all the other variables remain the same".
>
> Here, we present a toy example, suppose we have two agents $A$ and $B$, and their reflectors are $X$ and $Y$, respectively. For $X$, we first run a task with $(A+X, B+Y)$ and obtain the reward $r(A+X, B+Y)$, and then we run the same task with $(A, B+Y)$ to get the reward $r(A, B+Y)$, then the counterfactual reward for $X$ is $r_X = r(A+X, B+Y) - r(A, B+Y)$. Similarly, the counterfactual reward for $Y$ is $r_Y = r(A+X, B+Y) - r(A+X, B)$.
>
> Since $r(A, B+Y)$ and $r(A+X, B)$ are very different, the counterfactual rewards for different agents are various. For example, in the HotPotQA dataset, after a failed trial, the student agent $A$ generates a reflection $X$: "The student did not find the correct answer due to incomplete search scope and insufficient specificity...", and the teacher agent $B$ generates a reflection $Y$: "I think the main reason for failure was the unclear pronoun usage...". We first add both the reflections to the corresponding agent's memory, and run the next trial to calculate the episode difference score $r(A+X, B+Y)=1.0$. Then we run the same task with $(A, B+Y)$ and $(A+X, B)$ respectively and obtain $r(A, B+Y)=0$ and $r(A+X, B)=0.211$. The counterfactual reward of $X$ and $Y$ can be calculated as follows:
> $$
> r_X = r(A+X, B+Y) - r(A, B+Y) = 1.0
> \notag
> $$
>
> $$
> r_Y = r(A+X, B+Y) - r(A+X, B) = 0.789
> \notag
> $$
>
>
> **Q3: With the complex PPO training, COPPER's performance is not very impressive, especially when the trial is small (in GSM8k and Checkmate in One Move of Figure 4 and Figure 8).**
>
> The trial number in these figures means that, for the same task, how many times the agents reflect on their behaviors to achieve better task success rates.
> We believe this could be a character of the RLHF method, that is, when using RLHF, we need to let the agents reflect more times to achieve better performances.
>
>
> **Q4: The left part of Figure 2 is a little confusing due to the position of Step 1 to 4. The execution order is unclear. The text in Figure 2 is too small to be seen, especially the right part.**
>
> The Step 1 to 4 on the left part of Figure 2 describes the process where the multi-agent system generates actions in response to problem $k$  at time $t$, which corresponds to lines 136-138 in Section 4.1. First, the system needs to compute the identifier $i$ of the agent to respond at the current time (Step 1). Then, agent $i$ updates its memory, including reflections of previous trials and the current trial's historical interactions (Step 2), and perceives the environmental state such as the question and current task scores (Step 3). Finally, the agent $i$ generates the action based on its own memory and the environment state (Step 4).
>
> To make Figure 2 more clear and accurate,  we will add explanations of steps 1 to 4, and replot the figure in the final version. The reploted version can be found in Figure 1 of the PDF.
>
>
> **Q5: Will there be any negative counterfactual reward? For example, the removal of a specific reflection will improve the performance.**
>
> Yes, sometimes removing the reflection of a certain agent can lead to an improvement in the task performance. To verify this, we further count the number of negative reflections in the constructed training data. The statistics are shown in Table 1 and Table 2 of the PDF.
>
> **Q6: What is the impact of the agent profile on the reflector? Will a personalized reflector be better than a general reflector?**
>
> Thanks for this very interesting question. Inspired by your comments, we conducted additional experiments by removing the personalized profiles. The results are presented in Figure 5 of the PDF.
>
> From the results, we can observe that when using pre-trained LMs (LongChat and GPT-3.5) to reflect, the removal of the agent profile has a greater impact on the GPT-3.5 reflector. This may be due to GPT-3.5's better contextual understanding ability. Our COPPER can further improve the model's reflection ability under non-profile settings. However, the results are slightly worse than the setting with agent profiles.

---

> > ### Comment · Reviewer_gdf9 · 2024-08-11
> >
> > Thanks for the author's detailed responses and I would like to keep my score.

---

> > > ### Author Response · Authors · 2024-08-12
> > >
> > > Thanks very much for your feedback. Your comments are very constructive, and we will incorporate them in the final version.

---

### Author Rebuttal · Authors · 2024-08-07

Dear reviewers:

Thanks for your detailed reviews. Additional tables and figures mentioned in the rebuttals are shown in the submitted one-page pdf.

---

### Decision · Program_Chairs · 2024-09-25

**Decision:**

Accept (poster)

**Comment:**

The paper proposes COPPER, a framework designed to enhance collaboration in multi-agent systems through a learnable self-reflection mechanism. The method leverages counterfactual rewards to address the credit assignment problem and uses a shared reflector to generate personalized reflections tailored to each agent's role. The approach has been evaluated across various tasks, including multi-hop question answering, mathematics, and chess, and is reported to show improvements over existing models.

The reviewers generally acknowledge the novelty of applying counterfactual rewards and the comprehensive experimental analysis provided in the paper. However, concerns were raised about the practical effectiveness of COPPER, particularly in scenarios where complete information sharing is unrealistic or where computational costs are a concern. The introduction of Retroformer as a baseline under a multi-agent setting was requested by one reviewer, and while the authors addressed this in their rebuttal, the improvement over Retroformer was deemed marginal, especially in early trials.

Another significant point is the motivation behind the shared reflector. The potential misalignment with real-world scenarios, where full information sharing is not feasible, was highlighted. The authors attempted to mitigate this concern by presenting additional experiments under partial information settings. The paper's focus on combining various computationally intensive methods also drew criticism, with one reviewer questioning whether the resulting performance gains justify the increased costs.

Overall, multi-agent collaboration is an emerging topic, and this paper can contribute to the field by providing certain insights on applying counterfactual rewards, and cost could be optimized in the future work.